# IN-PLACE TEST-TIME TRAINING

**Guhao Feng**[1,2*]   **Shengjie Luo**[1*]   **Kai Hua**[1]   **Ge Zhang**[1]
**Wenhao Huang**[1†]   **Di He**[2†]   **Tianle Cai**[1]

[1]ByteDance Seed
[2]State Key Laboratory of General Artificial Intelligence, Peking University

## ABSTRACT

The static "train then deploy" paradigm fundamentally limits Large Language Models (LLMs) from dynamically adapting their weights in response to continuous streams of new information inherent in real-world tasks. Test-Time Training (TTT) offers a compelling alternative by updating a subset of model parameters (fast weights) at inference time, yet its potential in the current LLM ecosystem is hindered by critical barriers including architectural incompatibility, computational inefficiency and misaligned fast weight objectives for language modeling. In this work, we introduce *In-Place Test-Time Training (In-Place TTT)*, a framework that seamlessly endows LLMs with Test-Time Training ability. In-Place TTT treats the final projection matrix of the ubiquitous MLP blocks as its adaptable fast weights, enabling a "drop-in" enhancement for LLMs without costly retraining from scratch. Furthermore, we replace TTT's generic reconstruction objective with a tailored, theoretically-grounded objective explicitly aligned with the Next-Token-Prediction task governing autoregressive language modeling. This principled objective, combined with an efficient chunk-wise update mechanism, results in a highly scalable algorithm compatible with context parallelism. Extensive experiments validate our framework's effectiveness: as an in-place enhancement, it enables a 4B-parameter model to achieve superior performance on tasks with contexts up to 128k, and when pretrained from scratch, it consistently outperforms competitive TTT-related approaches. Ablation study results further provide deeper insights on our design choices. Collectively, our results establish In-Place TTT as a promising step towards a paradigm of continual learning in LLMs.

## 1   INTRODUCTION

Large Language Models (LLMs) have demonstrated remarkable capabilities across a range of complex tasks (Brown et al., 2020; Chowdhery & et al., 2022; Touvron et al., 2023; OpenAI, 2024). This success is largely built on a static "train then deploy" paradigm, where a model first acquires knowledge from massive corpora and then kept fixed during inference. Yet this design imposes a fundamental limitation: once deployed, the model's weights cannot be updated, preventing dynamic adaptation to the specific context provided by streaming input tokens. Consequently, at test time, the model is constrained in its ability to process and reason over long-horizon, evolving tasks (Chan et al., 2024; Starace et al., 2025), and to continuously learn from unbounded streams of experience like humans (Silver & Sutton, 2025).

In-context learning (Brown et al., 2020; Wei et al., 2023) offers a way to mitigate this problem via maintaining all past input tokens in the context. However, its effectiveness is tethered to the model's context window, restricted by the quadratic complexity of the de facto attention mechanism (Vaswani et al., 2017). This bottleneck has spurred a line of research into architectural solutions aimed at efficiently extending the context window (Beltagy et al., 2020; Peng et al., 2023; Child et al., 2019; Dao et al., 2023). Differently, *Test-Time Training (TTT)* has emerged as a new paradigm (Sun et al., 2020; Wang et al., 2021; Sun et al., 2024; Behrouz et al., 2024; Yau et al., 2025). Instead of merely

---

*Equal contribution.
†Correspondence to:
    Di He <dihe@pku.edu.cn> and Wenhao Huang <huang.wenhao@bytedance.com>.

making a static model more efficient, TTT enables the model to dynamically update the parameters and adapt to any specific context, directly targeting the aforementioned limitation. Specifically, TTT introduces a small subset of model parameters, called fast weights (Schlag et al., 2021), which can be updated on the fly for each new input. By minimizing a self-supervised reconstruction objective, these fast weights compress and internalize contextual information, functioning as an expressive, online evolving state.

Despite its conceptual appeal, unleashing TTT's potential within the current LLM ecosystem is hindered by critical barriers: (i) Existing TTT methods often rely on specialized layers beyond standard Transformer blocks, which usually demand costly pretraining from scratch to achieve satisfactory performance. (Sun et al., 2020; Wang et al., 2021; Zhang et al., 2025; Sun et al., 2024); (ii) the canonical TTT mechanism is inherently sequential (Sun et al., 2020; 2024). While existing works explore chunk-wise acceleration (Sun et al., 2023; Behrouz et al., 2024; Irie & Gershman, 2025; Yau et al., 2025), TTT's role as the primary token mixer forces a reliance on small chunks to maintain performance, thereby bottlenecking the massive parallelism required to saturate modern accelerators; and (iii) the prevalent use of a generic reconstruction objective for TTT's fast weights updating is not explicitly tailored for the causal, Next-Token Prediction task that governs autoregressive LMs, potentially hindering their ultimate performance.

To bridge this gap, we introduce *In-Place Test-Time Training (In-Place TTT)*, a framework designed to seamlessly endow LLMs with Test-Time Training capabilities by directly addressing the afore-mentioned barriers. Our core insight is to repurpose existing MLP blocks with an in-place design rather than introducing a new, specialized layer (tackling barrier i). Specifically, In-Place TTT treats the final projection matrix of MLP blocks as their fast weights, updating it in-place during inference. This "drop-in" design requires no modifications to the model's architecture, preserving the integrity of pre-trained weights and enabling on-the-fly adaptation without costly retraining from scratch.

To tackle the computational inefficiency and objective misalignment, we further design a bespoke adaptation mechanism for language modeling. Following previous works (Sun et al., 2023; Behrouz et al., 2024; Irie & Gershman, 2025; Zhang et al., 2025; Yau et al., 2025), we replace the inefficient per-token updates with a scalable chunk-wise update rule (tackling barrier ii). Furthermore, our in-place design operates complementarily to the attention mechanism. This synergy obviates the need for small chunks required by standalone TTT layers, thereby ensuring high throughput on modern accelerators. Concurrently, we move beyond the generic reconstruction targets of prior work (Sun et al., 2024; Zhang et al., 2025) and introduce a novel objective explicitly aligned with the Next-Token Prediction (NTP) goal (tackling barrier iii). Grounded in a rigorous theoretical analysis, we show this NTP-aligned objective encourages the fast weights to store predictively useful information for autoregressive language modeling, leading to a highly effective and scalable algorithm.

Grounded in these principled design choices, our In-Place TTT provides a practical and effective framework for enhancing LLMs with dynamic, continual adaptation. We conduct extensive experiments on language modeling tasks of various compute scales, using them as a practical proxy to probe the model's potential on long-horizon, evolving tasks. Through relatively cheap continual training, our In-Place TTT enables Qwen3-4B-Base to achieve superior performance on tasks with contexts up to 128k. Furthermore, we compare In-Place TTT with competitive TTT-related methods by conducting pretraining from scratch on up to 32k-length corpora, validating the architectural merit of our framework. Finally, ablation studies on state size, chunk size, and fast weight objectives provide deeper insights, confirming the critical role of each design choice. Collectively, our results establish In-Place TTT as a promising step towards a paradigm of continual learning in LLMs.

## 2 PRELIMINARY: TEST-TIME TRAINING

This section introduces *Test-Time Training (TTT)*, a paradigm that enables models to adapt dynamically to new data at inference time (Sun et al., 2020; 2024; Zhang et al., 2025). We will first elaborate on the TTT mechanism and then discuss the key desiderata for successfully applying TTT to LLMs, which directly motivates our framework.

**The TTT mechanism.** At its core, the TTT mechanism leverages *fast weights* (Ba et al., 2016; Schlag et al., 2021), denoted by $\mathbf{W}$. These weights constitute a small neural network $f_{\mathbf{W}}(\cdot) : \mathbb{R}^d \to \mathbb{R}^d$, which is rapidly updated at test time. Unlike standard model weights that are frozen after training, the

fast weights $\mathbf{W}$ act as a dynamic memory, continuously storing and retrieving contextual information from the sequence.

To process an input sequence $\mathbf{x} = [x_1, x_2, \ldots, x_N]$, each token $x_i \in \mathbb{R}^d$ is typically projected to derive the necessary inputs for the TTT operations, such as a *query* ($q_i$), a *key* ($k_i$), and a *value* ($v_i$). The TTT mechanism then operates through two core, sequential operations:

1. **Update Operation:** The fast weights $\mathbf{W}$ are updated to associate a key $k_i$ with its corresponding value $v_i$. This is framed as a single optimization step that minimizes a loss function $\mathcal{L}(\cdot, \cdot)$ (e.g., Mean Squared Error), which measures the discrepancy in this association. Intuitively, this step encodes the information from the $(k_i, v_i)$ pair into the neural memory $f_{\mathbf{W}}$. Given a learning rate $\eta$, the update rule is:

$$\mathbf{W}_i \leftarrow \mathbf{W}_{i-1} - \eta \nabla_{\mathbf{W}} \mathcal{L}\big(f_{\mathbf{W}_{i-1}}(k_i), v_i\big).$$

2. **Apply Operation:** The newly updated network $f_{\mathbf{W}}$, now parameterized by $\mathbf{W}_i$, is used to process a query $q_i$, i.e., $o_i = f_{\mathbf{W}_i}(q_i)$. This output $o_i$ is enriched with the contextual information from preceding key-value pairs, as that information is now encoded in $\mathbf{W}_i$.

While this two-step formulation describes the high-level mechanism of TTT, the specific implementation details can vary significantly. Indeed, numerous recent studies have investigated a rich design space, exploring different loss functions, more sophisticated optimizers, and alternative neural memory parameterizations to improve performance and efficiency (Wang et al., 2025; Behrouz et al., 2024; 2025b; Karami & Mirrokni, 2025). These design choices critically influence how effectively the fast weights can store, retrieve, and forget sequential information, positioning the TTT mechanism for different data modalities and tasks.

**Desiderata for TTT within the LLM ecosystem.** Despite its promise as a paradigm for dynamic adaptation, unleashing TTT's potential within the LLM ecosystem requires addressing several critical challenges. For TTT to be a viable and effective component, it must satisfy the following desiderata:

- **Architectural Compatibility**. We call an architecture compatible with LLM if it can warm start from a pretrained checkpoint. However, current TTT mechanisms are often developed as standalone recurrent layers designed to replace attention, rather than complement it (Sun et al., 2020; Wang et al., 2021; Zhang et al., 2025; Sun et al., 2024; Hu et al., 2025). This necessitates costly pretraining from scratch, creating a significant barrier to adoption for the massive, billion-parameter models that dominate the LLM ecosystem. Therefore, a key desideratum is a "drop-in" design that requires no fundamental architectural modifications.

- **Computational Efficiency**. The mechanism must be efficient on modern parallel accelerators. The canonical per-token update rule of TTT is inherently sequential and, as a result, severely bottlenecks the parallel processing capabilities of GPUs and TPUs (Sun et al., 2020; 2024; Behrouz et al., 2024). This operational inefficiency makes fine-grained updates impractical for high-throughput language modeling. Consequently, an efficient TTT implementation must move beyond per-token schemes and ensure scalability, for instance by adopting chunk-wise update mechanisms (Li et al., 2025; Sun et al., 2023; Behrouz et al., 2024; Irie & Gershman, 2025).

- **Tailored Learning Objective for Language Modeling**. The predominant self-supervised objective in TTT is reconstruction, where the model learns to associate $(k_i, v_i)$ pairs, and $v_i$ is typically derived from the input token $x_i$ itself (Sun et al., 2020; 2024; Zhang et al., 2025; Wang et al., 2021; Hu et al., 2025). While this generic objective enables the TTT mechanism to store information, its direct relevance to the ultimate goal of language modeling—predicting the next token—is not guaranteed. The choice of the target value $v$ remains a critical, yet underexplored, design decision that may be suboptimal for capturing the complex causal dependencies required for LLMs.

## 3 IN-PLACE TEST-TIME TRAINING

To satisfy the desiderata outlined in Section 2, we introduce ***In-Place Test-Time Training (In-Place TTT)***, a framework designed to unlock TTT capabilities for LLMs. We first present our overall

framework, which resolves architectural incompatibility via an in-place design that repurposes existing MLP blocks, while ensuring computational efficiency with a chunk-wise update mechanism (Section 3.1). We then detail our novel LM-aligned objective, which is explicitly designed for LLMs by aligning with the Next-Token Prediction (NTP) goal (Section 3.2). Following this, we provide a theoretical analysis of our objective's superior properties (Section 3.3) and conclude with practical implementation details (Section 3.4).

## 3.1 OVERALL FRAMEWORK

**Repurposing MLP Blocks for In-Place Adaptation.** Previous TTT research has largely positioned it as a potential solution to replace the attention mechanism. However, these prior studies were typically conducted at moderate scales, a regime vastly different from that of modern, billion-parameter LLMs. Consequently, replacing the core attention mechanism—whose learned properties are critical to an LLM's capabilities—is a high-risk architectural modification. Moreover, introducing any new, randomly-initialized layer also creates a conflict with the billions of trained parameters of LLMs, necessitating costly and often impractical retraining to resolve this imbalance.

Our core insight is to sidestep these challenges entirely. Instead of replacing or adding components, we repurpose a ubiquitous module–the Multi-Layer Perceptron (MLP) block–to also serve as the fast weights. Recalling the TTT formulations in Section 2, there exist no constraints on the choice of fast weights, i.e., any parameters can serve as fast weights updated via the TTT mechanism. In particular, the MLP blocks in Transformers can also be viewed as a form of key-value memory (Geva et al., 2020), functioning as a "slow weights" for the vast, general knowledge acquired during pre-training. It is therefore a natural extension to leverage this same component to also function as the adaptive "fast weights", dynamically internalizing transient, in-context information at inference time.

Formally, we adapt the widely used gated MLP architecture (Grattafiori et al., 2024; Yang et al., 2025). Given the hidden representation $\mathbf{H}$, the gated MLP computes its output representation $\mathbf{O} = \left( \phi(\mathbf{H}\mathbf{W}_{\text{gate}}^{\top}) \odot (\mathbf{H}\mathbf{W}_{\text{up}}^{\top}) \right) \mathbf{W}_{\text{down}}^{\top}$. In our framework, we treat the input projections $\mathbf{W}_{\text{up}}$ and $\mathbf{W}_{\text{gate}}$ as frozen slow weights, while repurposing the final projection matrix, $\mathbf{W}_{\text{down}}$, as the adaptable fast weights. By exclusively updating $\mathbf{W}_{\text{down}}$ in-place, we preserve the model's architectural integrity, transforming TTT from a disruptive restructuring into a lightweight, "drop-in" enhancement for LLMs.

**Efficient Adaptation with Chunk-Wise Updates.** Beyond architectural compatibility, our in-place design also unlocks significant computational efficiencies. Conventional TTT methods, by aiming to replace the attention mechanism, were bound to inefficient per-token updates to enforce strict causality and perform fine-grained token mixing. Chunk-wise updating approaches have been explored by recent works to achieve acceleration (Li et al., 2025; Sun et al., 2023; Behrouz et al., 2024; Irie & Gershman, 2025). Our framework also follows to sidestep the trade-off entirely. Since we only adapt the MLP blocks and leave the attention layers intact, we are liberated from the per-token constraint, enabling a far more efficient chunk-wise update strategy which further bypasses the small chunk constraints (our ablation study results (Section 4.3) also verify that our framework is naturally well-suited for chunk-wise—and specifically large chunk-wise—updates, achieving optimal performance with chunk sizes of 512 to 1024.

The process operates as follows. Given the intermediate activations $\mathbf{Z} = \phi(\mathbf{H}\mathbf{W}_{\text{gate}}^{\top}) \odot (\mathbf{H}\mathbf{W}_{\text{up}}^{\top}) \in \mathbb{R}^{n \times d_{\text{ff}}}$ and corresponding value targets and outputs $\mathbf{V}, \mathbf{O} \in \mathbb{R}^{n \times d_{\text{model}}}$, we partition them into $k$ non-overlapping chunks of size $C$, denoted $\square_{[i]} = \square_{iC+1:(i+1)C} \in \mathbb{R}^{C \times d'}$ for $\square \in \{\mathbf{Z}, \mathbf{V}, \mathbf{O}\}, i \in [k]$ and $d'$ being their corresponding dimension. Let $\mathbf{W}_{\text{down}}^{(i)}$ be the fast weights state before processing chunk $i$ and $\mathbf{W}_{\text{down}}^{(0)} = \mathbf{W}_{\text{down}}$. For each chunk $i \in [k]$, we perform two sequential operations:

1. **Apply Operation:** The current state of the fast weights $\mathbf{W}_{\text{down}}^{(i)}$ are used to process chunk $\mathbf{Z}_{[i]}$, i.e., $\mathbf{O}_{[i]} = \mathbf{Z}_{[i]}(\mathbf{W}_{\text{down}}^{(i)})^{\top}$.

2. **Update Operation:** The fast weight $\mathbf{W}_{\text{down}}^{(i)}$ are updated using $\mathbf{Z}_{[i]}$ as keys and $\mathbf{V}_{[i]}$ as values, which is performed via one gradient descent step with a loss function $\mathcal{L}$ and a learning rate $\eta$: $\mathbf{W}_{\text{down}}^{(i+1)} = \mathbf{W}_{\text{down}}^{(i)} - \eta \nabla_{\mathbf{W}} \mathcal{L} \left( \mathbf{Z}_{[i]}(\mathbf{W}_{\text{down}}^{(i)})^{\top}, \mathbf{V}_{[i]} \right)$.

This chunk-wise update strategy is designed for modern hardware, similar to prior attempts (Li et al., 2025; Sun et al., 2023; Behrouz et al., 2024; Irie & Gershman, 2025). Moreover, due to our in-place MLP adaptation, we can use a large chunk size $C$ to process large blocks of tokens at once, thereby highly leveraging parallelism and utilizing the computational power of GPUs or TPUs.

## 3.2 LM-ALIGNED OBJECTIVE

With the efficient, in-place adaptation framework established, the performance of In-Place TTT now hinges on the design of its learning objective. In this subsection, we introduce our Language Modeling-Aligned objective, which is explicitly tailored for LLMs.

Prior TTT approaches typically use a reconstruction target, e.g., $\mathcal{L}(f_{\mathbf{W}}(k), v)$ where both $k$ and $v$ are linear projection outputs of the same input token $x$ (Sun et al., 2020; 2024; Zhang et al., 2025), which encourages the model to simply memorize the current token's representation. We argue that this is suboptimal for language modeling tasks. Instead, we propose to align the objective with the Next-Token Prediction (NTP) goal governing LLMs.

To achieve this, we specify the target $v$ to include future token information. Formally, we derive our target $\hat{\mathbf{V}} = \mathrm{Conv1D}(\mathbf{X}_0)\mathbf{W}_{\mathrm{target}}$, where $\mathbf{X}_0 \in \mathbb{R}^{n \times d_{\mathrm{model}}}$ denotes the token embedding, $\mathrm{Conv1D}(\cdot)$ is the 1D Convolution operator and $\mathbf{W}_{\mathrm{target}} \in \mathbb{R}^{d_{\mathrm{model}} \times d_{\mathrm{model}}}$ is a trainable projection matrix. Under this formulation, the amount of future token information can be controlled in our target $\hat{\mathbf{V}}$, e.g., the Next-Token target can be achieved by parameterizing $\mathbf{W}_{\mathrm{target}}$ as an identity transformation and assigning $\mathrm{Conv1D}(\cdot)$'s kernel weights to be 1 for the next token and 0 for other tokens.

With this aligned target, we use the widely used similarity measure to instantiate our loss function for simplicity, i.e., $\mathcal{L}(\cdot, \cdot) = -\langle \cdot, \cdot \rangle_F$. Under this loss function, the gradient with respect to the fast weights in our chunk-wise mechanism can be directly derived:

$$\mathbf{W}_{\mathrm{down}}^{(i)} = \mathbf{W}_{\mathrm{down}}^{(i-1)} + \eta \hat{\mathbf{V}}_{[i]}^{\top} \mathbf{Z}_{[i]}. \tag{1}$$

## 3.3 THEORETICAL ANALYSIS

Intuitively, our LM-Aligned objective explicitly encourages the fast weights to compress predictively useful information for future tokens, thereby enhancing the model's capacity for dynamic adaptation. In this subsection, we formalize this intuition by theoretically analyzing the benefits of our objective. We ground our analysis within the canonical *induction head* setting (Olsson et al., 2022; Elhage et al., 2021), a mechanism understood to be critical for in-context learning in LLMs.

**Setup.** Consider an input sequence where a key-value pair, $(x_{t^*}, x_{t^*+1}) = (k^*, v^*)$, appears at an arbitrary position $t^*$. Subsequently, at a query position $n > t^*$, the key $k^*$ reappears, such that $x_n = k^*$. The model must then correctly predict the associated value, $x_{n+1} = v^*$.

Without loss of generality, we analyze a single Transformer block enhanced by our In-Place TTT. Let $\mathbf{Z}_t \in \mathbb{R}^{d_{\mathrm{ff}}}$ be the intermediate activation of token $x_t$ and $\mathbf{E}_w \in \mathbb{R}^{d_{\mathrm{model}}}$ be the token embedding for $x_w$. In our framework, the fast weights update from prior context chunks is $\Delta \mathbf{W}_{\mathrm{down}} = \eta \sum_{t \in \mathrm{prior\ chunks}} \mathbf{V}_t \mathbf{Z}_t^{\top}$. This update then change the output logit at the query position $n$ by $\Delta \ell_n[w] = \mathbf{E}_w^{\top} \Delta \mathbf{W}_{\mathrm{down}} \mathbf{Z}_n$. We compare two choices for the TTT target $\mathbf{V}_t$:

- *Reconstruction Target:* $\mathbf{V}_t^{\mathrm{rec}} = \mathbf{E}_{x_t}$, the embedding of the *current* token.
- *LM-Aligned Target:* $\mathbf{V}_t^{\mathrm{LM}} = \mathbf{E}_{x_{t+1}}$, the embedding of the *next* token.

**Assumptions.** Our analysis rests on two mild assumptions about the properties of the embeddings and intermediate activations, which are standard in theoretical analyses of Transformers:

1. *Approximate Orthogonality of Embeddings*: For any two distinct tokens $w_i, w_j \in \mathcal{V}$, their embeddings are nearly orthogonal: $|\mathbf{e}_{w_i}^{\top} \mathbf{e}_{w_j}| \le \epsilon$ for a small constant $\epsilon > 0$. Additionally, embeddings have a non-trivial magnitude: $\|\mathbf{e}_{w_i}\|^2 \ge c_{\mathrm{norm}}^2 > 0$ for some constant $c_{\mathrm{norm}}$.
2. *Key-Query Alignment*: The intermediate activations $\mathbf{Z}_n$ for the query token $x_n = k^*$ is aligned with $\mathbf{Z}_{t^*}$ of its corresponding key token $x_{t^*} = k^*$: $\mathbb{E}[\mathbf{z}_{t^*}^{\top} \mathbf{z}_n] = c_{\mathrm{align}} > 0$. For other positions $t \ne t^*$, the tokens are unrelated to the query, i.e, $\mathbb{E}[(\mathbf{V}_t \mathbf{Z}_t^{\top}) \mathbf{Z}_n] = \mathbf{0}$.

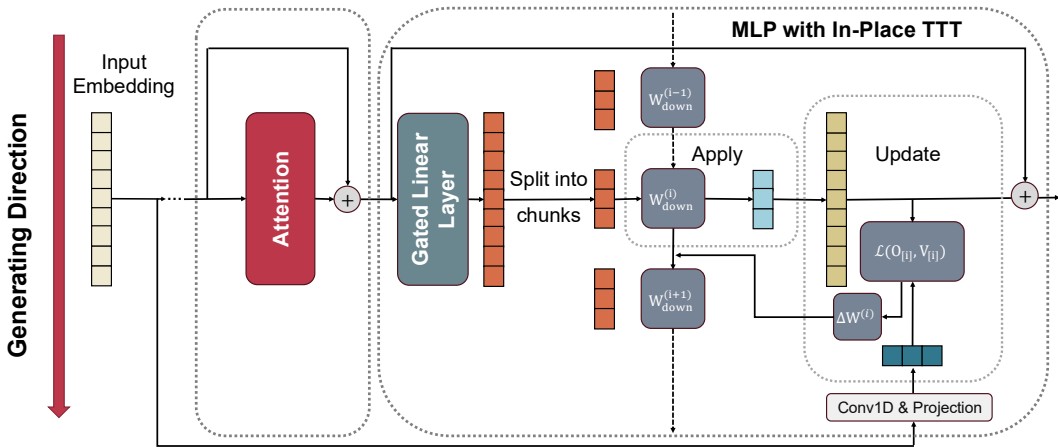

Figure 1: The overall framework of our In-Place Test-Time Training. The module operates sequentially on input chunks. For each chunk, the current fast weights are first *applied* to the intermediate activations $\mathbf{Z}$ to produce the output. Then, these weights are *updated* using the activations $\mathbf{Z}$ and a value $\mathbf{V}$ derived from the token embeddings. This "apply-then-update" cycle allows the model to dynamically adapt to incoming context in a strictly causal manner.

With this setup, we present our main theoretical result:

**Theorem 1** (Logit-wise Effect of LM-Aligned Target v.s. Reconstruction Target). *Under the specified setup and assumptions, for a learning rate $\lambda_{lr} > 0$, the expected change in logits $\Delta\ell_n$ after one update step using the* LM-Aligned target *satisfies:*

$$\text{(Correct logit increases)} \quad \mathbb{E}\left[\Delta\ell_n[v^*]\right] \geq \lambda_{lr} \cdot c_{norm}^2 \cdot c_{align}, \tag{2}$$

$$\text{(Other logits almost unchanged)} \quad |\mathbb{E}\left[\Delta\ell_n[w]\right]| \leq \lambda_{lr} \cdot \epsilon \cdot c_{align}, \quad \forall w \neq v^*. \tag{3}$$

*In contrast, for the* reconstruction target, *the expected change in logits is negligible for the correct token:* $|\mathbb{E}\left[\Delta\ell_n[v^*]\right]| \leq \lambda_{lr} \cdot \epsilon \cdot c_{align}.$

The proof is provided in Appendix B. In Theorem 1, the LM-Aligned target is guaranteed in expectation to increase the logit of the correct next token $v^*$ and keep that of other tokens approximately unchanged, directly aiding the model's prediction task. In contrast, the reconstruction target provides no such predictive benefit, failing to increase the logit of the correct token. In practice, our implementation extends this principle from a single next token to a learned, localized combination of future tokens, which also aligns with recent promising results of Multi-Token Prediction in advanced LLMs (Liu et al., 2024) as an effective extension of the NTP objective. This allows our In-Place TTT to capture a richer predictive signal, thereby compressing useful contextual information more effectively than simple reconstruction.

### 3.4 IMPLEMENTATION DETAILS

Combining aforementioned designs, Figure 1 illustrates our In-Place TTT framework. Here we further elaborate on practical implementation details, which are engineered for high efficiency and scalability on modern hardware. In particular, our approach is fully compatible with *Context Parallelism (CP)*, relying on a parallel scan algorithm to process sequence chunks simultaneously while preserving the strict causal semantics of an auto-regressive update. Additional discussions are further presented.

**Efficient Implementation with Context Parallelism.** The associative nature of our update rule in Equation (1) makes In-Place TTT amenable to a context-parallel implementation, which partitions a sequence along its length and processes the chunks simultaneously. The process unfolds into three stages: (i) for all chunks $i \in \{1, \ldots, T\}$, we compute the intermediate activations $\mathbf{Z}_{[i]}$ and the fast weight update $\Delta\mathbf{W}_{down}^{(i)} = (\hat{\mathbf{V}}_{[i]})^\top \mathbf{Z}_{[i]}$ in parallel; (ii) a single prefix sum over $[..., \Delta\mathbf{W}_{down}^{(i)}, \Delta\mathbf{W}_{down}^{(i+1)}, ...]$ is conducted to compute the aggregated updates for each chunk: $\Delta\mathbf{S}_i = \sum_{j=1}^{i-1} \Delta\mathbf{W}_j$, which can be highly efficient on modern accelerators; (iii) the ef-

fective fast weights for each chunk, $\mathbf{W}_{\text{down}}^{(i-1)} = \mathbf{W}_{\text{down}}^{(0)} + \eta \Delta \mathbf{S}_i$, and the corresponding output, $\mathbf{O}_{[i]} = \mathbf{Z}_{[i]}(W_{\text{down}}^{(i-1)})^\top$, are computed in parallel.

**Causality and Boundary Handling.** To ensure that the update delta for chunk $i$ itself contains no future information, we apply causal padding to the 1D convolution when generating the value. This isolates each delta calculation to its respective chunk, making the parallel scan mathematically equivalent to a sequential update. Moreover, at document boundaries, the fast weights are reset to their pre-trained state to prevent context leakage across independent sequences. The final context parallel algorithm is presented in Algorithm 1 in Appendix C.

**Discussion.** In summary, our implementation of In-Place TTT synergistically combines a simple, computationally efficient update rule with a parallel scan algorithm. This design choice makes our method not only fast and scalable but also mathematically equivalent to a strictly causal sequential process, thanks to careful boundary and padding management. The resulting module is *CP-native*, fully causal, and can be seamlessly integrated as a drop-in replacement for the MLP block in standard Transformer architectures. Lastly, it is also noteworthy that the core principles of our framework are orthogonal to the specific choice of loss functions and its optimizer, which have been widely studied in the broader TTT literature, an exploration we leave as a promising direction for future work.

## 4 EXPERIMENTS

In this section, we conduct a series of experiments to empirically validate the effectiveness of our In-Place TTT framework. Specifically, we aim to answer the following research questions:

- **Q1**: How effectively can In-Place TTT enhance pre-trained LLMs in a "drop-in" manner?
- **Q2**: When trained from scratch, how does In-Place TTT compare against prior TTT approaches?
- **Q3**: What are the effects of key design choices in our In-Place TTT framework?

Using language modeling tasks of various scales as a practical proxy, we answer each question with carefully designed experiments in the following sub-sections. Due to space limits, we present detailed descriptions of experimental settings in Appendix D.

| Model | In-Domain Evaluation | | | | | | Extrapolation |
|---|---|---|---|---|---|---|---|
| | 4k | 8k | 16k | 32k | 64k | 128k | 256k |
| Mistral-7B (Jiang et al., 2023) | 93.6 | 91.2 | 87.2 | 75.4 | 49.0 | 13.8 | - |
| GLM3-6B (GLM, 2024) | 87.8 | 83.4 | 78.6 | 69.9 | 56.0 | 42.0 | - |
| Phi3-medium-14B (Abdin et al., 2024) | 93.3 | 93.2 | 91.1 | 86.8 | 78.6 | 46.1 | - |
| Llama3-8B (Pekelis et al., 2024) | 92.8 | 90.3 | 85.7 | 79.9 | 76.3 | 69.5 | - |
| Qwen3-4B (Instruct) (Yang et al., 2025) | 95.1 | 93.6 | 91.0 | 87.8 | 77.8 | 66.0 | - |
| Baseline | **96.6** | 94.1 | 92.1 | 88.7 | 74.3 | 74.8 | 41.7 |
| In-Place TTT | 96.1 | **95.6** | **92.7** | **89.3** | **78.7** | **77.0** | **43.9** |

Table 1: Evaluation results on the RULER benchmark (Hsieh et al., 2024). We report the average accuracy (%) as scores, with the best results in **bold**.

### 4.1 IN-PLACE TTT AS A DROP-IN ENHANCEMENT FOR PRE-TRAINED LLMS

To validate In-Place TTT as a "drop-in" enhancement for existing, pre-trained LLMs, we start with the competitive open-sourced *Qwen3-4B-Base* model. Its original context window is 32k, thereby we can simulate the long-horizon, evolving tasks requiring Test-Time Training capabilities by language modeling tasks of varying context lengths. In particular, we compare the performance of (1) *Qwen3-4B-Base* (Baseline); (2) *Qwen3-4B-Base + In-Place TTT* (In-Place TTT). Both models undergo the exact same continual training curriculum, ensuring a fair comparison where our In-Place TTT is the only variable.

Table 2: Extension of In-Place TTT to LLaMA-3.1-8B and Qwen3-14B-Base on the RULER benchmark. We report the average accuracy (%) with the best results in **bold**.

| Base Model | Method | 4k | 8k | 16k | 32k | 64k | 64k+YaRN |
|---|---|---|---|---|---|---|---|
| LLaMA-3.1-8B | Baseline | 93.9 | 92.1 | 92.5 | 91.1 | 81.6 | – |
| | In-Place TTT | **94.4** | **93.0** | **93.3** | **91.7** | **83.7** | – |
| Qwen3-14B | Baseline | 96.8 | 95.0 | 94.6 | 90.7 | 67.9 | 81.3 |
| | In-Place TTT | **97.2** | **95.7** | **95.2** | **91.2** | **70.6** | **82.5** |

**Training and Evaluation.** The continual training curriculum is divided into two stages: an initial phase of ∼20B tokens with 32k context length, followed by a second phase of ∼15B tokens with 128k context length. The detailed descriptions of training dataset can be found in Appendix D.1. To effectively manage these long sequences, we adapt the model's Rotary Position Embeddings using YaRN (Peng et al., 2023). We evaluate the long-context performance of both models on the RULER benchmark (Hsieh et al., 2024) using the popular OpenCompass framework (Contributors, 2023), with context lengths ranging from 4k to 256k. The 256k setting specifically measures the models' ability to extrapolate beyond the 128k context length limit. Detailed descriptions of training details can be found in Appendix D.2.

**Results and Discussion.** The results, summarized in Table 1, demonstrate that In-Place TTT significantly boosts the long-context proficiency of the pre-trained model. In particular, a clear trend can be easily seen from the results: while both models are competitive at short contexts, Qwen3-4B-Base enhanced by our In-Place TTT establishes a consistent and widening advantage as the sequence length increases. It achieves substantial gains at the 64k and 128k context lengths. Crucially, this advantage is maintained when extrapolating to a 256k context, demonstrating superior generalization. These findings confirm that In-Place TTT can be seamlessly integrated into a pre-trained LLM to boost its long-context proficiency. The model's strong performance at and beyond the context length validates our method as a practical and powerful tool for extending the capabilities of existing LLMs.

**Extension to More Models.** To verify the generality of our approach, we further apply In-Place TTT as a drop-in enhancement to two additional models: LLaMA-3.1-8B (Grattafiori et al., 2024) and Qwen3-14B-Base (Yang et al., 2025), following the same continual training protocol. As shown in Table 2, In-Place TTT consistently improves the RULER scores across all context lengths on both models. The gains are particularly pronounced at longer contexts, e.g., +2.1 at 64k for LLaMA-3.1-8B and +2.7 at 64k for Qwen3-14B-Base. Moreover, the improvement is maintained when combined with YaRN (Peng et al., 2023) for position extrapolation, confirming that our method is orthogonal to RoPE extension techniques. These results, spanning different model families and scales (4B–14B), reinforce that In-Place TTT is a broadly applicable drop-in enhancement for pre-trained LLMs.

## 4.2 PRE-TRAINING FROM SCRATCH: A COMPARATIVE ANALYSIS

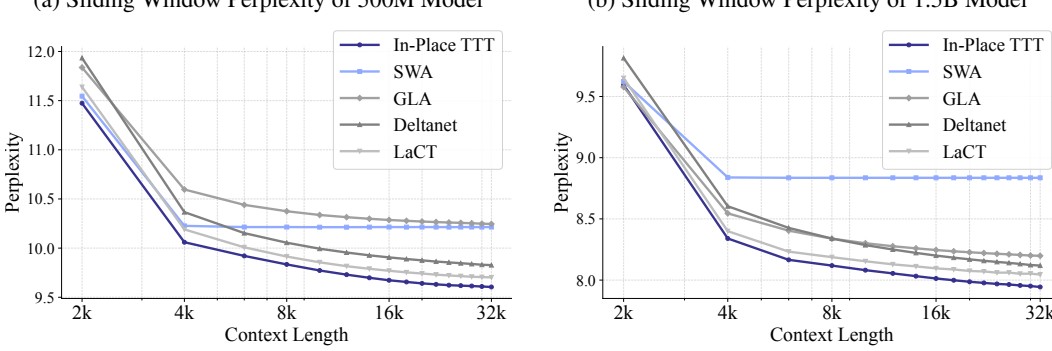

Figure 2: Sliding Window Perplexity at varying context lengths on the Pile dataset for 500M (left) and 1.5B (right) parameter models. Our In-Place TTT consistently achieves lower perplexity than all competitive baselines.

| Model Architecture | | Common Sense Reasoning | | | | | Long-Context Evaluation | | |
|---|---|---|---|---|---|---|---|---|---|
| | | HellaSwag | ARC-E | ARC-C | MMLU | PIQA | RULER-4k | RULER-8k | RULER-16k |
| Baselines | Full Attn. | 55.67 | 64.52 | 33.19 | 36.43 | 72.63 | 45.77 | 38.09 | 6.58 |
| | SWA | 54.92 | 64.18 | 32.85 | 36.06 | 72.58 | 14.77 | 9.91 | 5.07 |
| I.P. TTT | Full Attn. | **55.85** | **64.98** | 32.34 | **37.42** | **73.29** | 49.98 | 43.82 | **19.99** |
| | SWA | 55.24 | 64.60 | **33.70** | 36.48 | 72.03 | 28.33 | 26.80 | 7.57 |

Table 3: Evaluation results of 4B models on common sense reasoning and long-context evaluation benchmarks. Best performance is in **bold**. "SWA" is Sliding-Window Attention, "Full Attn." is Full Attention, and "I.P. TTT" is our In-Place TTT.

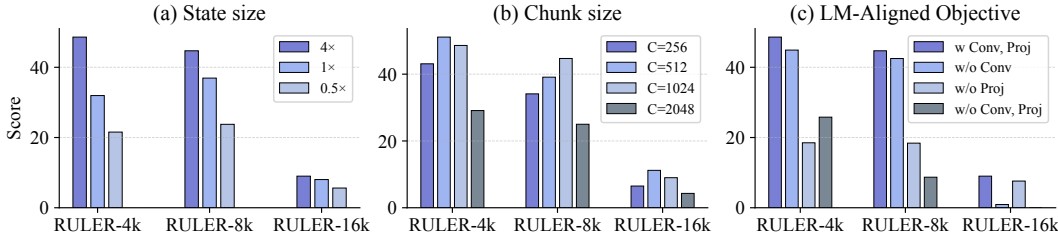

Figure 3: Ablation studies on the key design choices of the In-Place TTT framework, evaluated on the RULER benchmark with a 1.7B parameter model. The plots illustrate the impact of: (a) State size, showing that performance improves as the state size scales; (b) Chunk size, demonstrating a performance trade-off where intermediate sizes (e.g., 512, 1024) are optimal; and (c) The LM-Aligned Value objective, confirming that both the convolution (w Conv) and the projection (w Proj) are crucial.

Having demonstrated In-Place TTT's effectiveness as a "drop-in" module, we further evaluate its performance and scalability when integrated into models pre-trained from scratch. Our analysis proceeds in two stages: we first establish its language modeling capabilities at the 500M and 1.5B scales, and then assess its scalability and impact on a larger 4B model.

**Experimental Setup.** Firstly, we benchmark our In-Place TTT against prior TTT-related approaches and efficient attention methods based on TogetherAI (2024) at 500M and 1.5B parameter scales. Various competitive baselines are compared: (1) standard Transformer with sliding window attention (SWA) (Child et al., 2019; Beltagy et al., 2020) (2) Gated Linear Attention (GLA) (Yang et al., 2024b); (3) DeltaNet (Schlag et al., 2021; Yang et al., 2024d;a) (4) Large Chunk Test-Time Training (LaCT) (Zhang et al., 2025). For a fair comparison, both In-Place TTT and LaCT are built upon an SWA backbone. All models are trained on sequences with a 32k context length.

Building on these results, we further scale up to 4B-parameter models to evaluate scalability of our In-Place TTT approach. In particular, we compare Transformers with Full Attention and Transformers with SWA against their counterparts enhanced by our In-Place TTT. These models are trained for 120B tokens with an 8k context length. Detailed descriptions of datasets, model configurations, and training procedures are available in Appendix D.1, D.3, and D.2.

**Evaluation.** For the 500M and 1.5B models, we evaluate their long-context utilization using *Sliding Window Perplexity* on a validation set comprised of Pile (Gao et al., 2020) and Proof-Pile-2 (Paster et al., 2023). This metric measures perplexity on a fixed final block of tokens when extending the preceding context, where a decreasing perplexity trend indicates effective context usage. For the 4B models, we conduct a broader evaluation on a suite of downstream tasks, including common sense reasoning benchmarks (HellaSwag (Zellers et al., 2019), ARC (Clark et al., 2018), MMLU (Hendrycks et al., 2021b;a), PIQA (Bisk et al., 2019)) and the long-context RULER benchmark (Hsieh et al., 2024).

**Results and Discussion.** In Figure 2, we plot the sliding window perplexity against context length for both 500M and 1.5B model. It can be easily seen that our In-Place TTT consistently achieves lower validation perplexity than all competitive baselines, with its performance steadily improving up to the full 32k context. This sustained improvement suggests its core mechanism successfully compresses and utilizes information from incoming context.

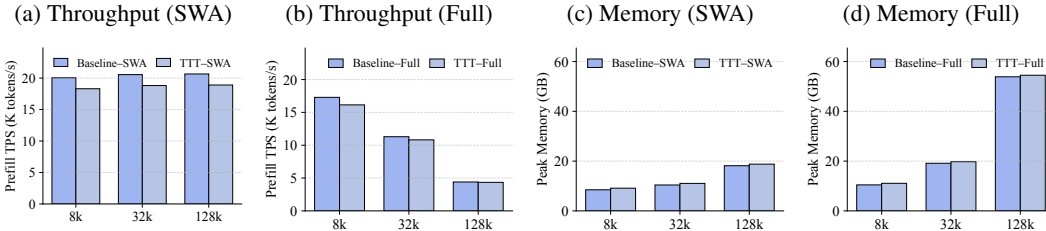

Figure 4: Efficiency analysis of In-Place TTT. Both prefill throughput (a, b) and peak memory (c, d) metrics are presented for 4B models with Sliding-Window Attention (SWA) and Full Attention at various context lengths. Our In-Place TTT introduces negligible overhead in practical scenarios.

Moreover, the results in Table 3 further show that 4B-parameter Transformers with both Full Attention and SWA are consistently improved across most common sense reasoning tasks. Furthermore, models with our In-Place TTT yield superior performance on the long-context evaluation, e.g., RULER-16k score is improved from 6.58 to 19.99 for the Transformer with Full Attention and RULER-8k score is boosted from 9.91 to 26.80 for the SWA model. These substantial gains, particularly across models of various scales, establish our In-Place TTT as a highly effective and scalable approach.

### 4.3 ABLATION STUDIES: ON THE IMPACT OF KEY DESIGN CHOICES

Lastly, we conduct a series of ablation studies on RULER with a 1.7B-parameter model, providing deeper insights into our design choices. Detailed settings are presented in Appendix D.3 and D.2.

**Impact of State Size.** We first investigate how performance scales with the fast weights size, which can be controlled by varying the number of TTT-enabled layers. Figure 3 (a) shows a clear trend that the performance of our In-Place TTT consistently improves along with the state size scaling. This confirms that larger fast weights allow the model to more effectively adapt to contextual information, which further supports our repurposing approach leveraging the large amount of MLP states.

**Impact of Chunk Size.** The chunk size $C$ in Section 3.1 controls both the granularity of fast weights updating and parallelism, exposing a tradeoff between efficiency and performance. By varying the chunk size, Figure 3 (b) shows that both $C = 512$ and $C = 1024$ competitively achieve better performance compared to other choices, while $C = 1024$ has better efficiency.

**Impact of LM-Aligned Objective.** Next, we delve deep into our tailored LM-Aligned objective, i.e., $\hat{\mathbf{V}} = \mathrm{Conv1D}(\mathbf{X}_0)\mathbf{W}_{\mathrm{target}}$ in Section 3.2. In particular, the $\mathrm{Conv1D}$ operator is used to yield targets containing future token information, and the $\mathbf{W}_{\mathrm{target}}$ is a projection transformation. In Figure 3 (c), we comprehensively ablate combinations of these components. The result shows that both of them are necessary for performance guarantee, while $\mathrm{Conv1D}$ plays an essential role on long context and $\mathbf{W}_{\mathrm{target}}$ is crucial on short context. These results align with our theoretical analysis in Section 3.3, strongly supporting our motivation to derive a tailored objective for language modeling.

**Efficiency Impact of In-Place TTT.** We further study the computational overhead introduced by our In-Place TTT. In Figure 4, we compare both prefill throughput and memory consumptions of with and without our In-Place TTT. The results indeed verify the efficiency of our practical implementations.

## 5 CONCLUSION

We introduced In-Place Test-Time Training, a practical framework that resolves the critical barriers of TTT for LLMs. Principled design choices are proposed including an in-place mechanism that repurposes existing MLP blocks, an efficient chunk-wise update rule, and a theoretically-grounded objective aligned with language modeling. Extensive experiments validate that our approach not only serves as a powerful "drop-in" enhancement for pre-trained LLMs but also outperforms strong baselines when trained from scratch. By providing a scalable solution for on-the-fly adaptation, our work makes a promising step towards a new paradigm of more dynamic, continual learning for LLMs.

## ETHICS STATEMENT

This work introduces a foundational model architecture and does not present any direct real-world applications with immediate ethical concerns. We acknowledge the broader societal risks associated with Large Language Models, such as potential biases inherited from training data, and advocate for further research into their responsible development and deployment.

## REPRODUCIBILITY STATEMENT

To ensure the reproducibility of our findings, we provide comprehensive details on all experimental settings, model configurations, and training hyperparameters in Appendix D. The theoretical claims are supported by a complete proof in Appendix B, and the implementation is detailed in the pseudocode in Appendix C. We will release source code and model checkpoints to facilitate further research.

## ACKNOWLEDGEMENT

DH is supported by National Science Foundation of China (NSFC62376007), National Science Foundation of China (under Key Project No. 92570203), Beijing Natural Science Foundation (Z250001) and Beijing Major Science and Technology Project under Contract no. Z251100008425004.

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

## A    RELATED WORK

**Test-Time Training (TTT).**    Test-Time Training (TTT) is a paradigm that enables a model to adapt dynamically in response to continuous streams of data at inference by updating a small subset of its parameters, known as *fast weights* (Ba et al., 2016). Initially demonstrating success in computer vision (Sun et al., 2020; Wang et al., 2021), TTT has since been extended to numerous other modalities—including language (Sun et al., 2024), video (Dalal et al., 2025), and audio (Dumpala et al., 2023)—underscoring its broad applicability. Research in this area has largely focused on two avenues for improving TTT's effectiveness: the design of more sophisticated test-time optimizers (Behrouz et al., 2025a) and the formulation of novel, self-supervised online learning objectives (Behrouz et al., 2024; Karami & Mirrokni, 2025). However, the computational efficiency of TTT remained a critical bottleneck due to its inherently sequential, per-token update process. The chunk wise Test-Time Training framework was the first to directly address this challenge by introducing a chunk-wise update mechanism to better leverage parallel hardware (Behrouz et al., 2024; Irie & Gershman, 2025; Zhang et al., 2025; Sun et al., 2023; Yau et al., 2025; Li et al., 2025). Despite these advances, TTT's function as the primary token mixer necessitates reliance on small chunks to preserve performance—this in turn creates a bottleneck that limits the massive parallelism needed to fully utilize modern accelerators. Furthermore, prior work has not addressed how to seamlessly integrate TTT into large, pre-trained models, nor developed learning objectives specifically tailored for the autoregressive nature of LLMs, which are gaps our work directly addresses.

**Efficient Long-Context Architectures.**    A parallel line of research seeks to extend the effective context window of LLMs by mitigating the quadratic complexity of the standard attention mechanism. Major approaches include: 1) Sparse attention methods, which restrict the range of token-to-token interactions via fixed patterns like sliding or strided windows (Child et al., 2019; Beltagy et al., 2020; Yuan et al., 2025); 2) Linear-time variants, which approximate the attention mechanism or replace it with efficient recurrent or gated formulations, such as linear attention (Katharopoulos et al., 2020; Schlag et al., 2021) Gated Linear Attention (GLA) (Yang et al., 2023); and 3) State-Space Models (SSMs), which compress sequence history into a compact latent state, enabling processing with linear complexity (Dao et al., 2023; Dao & Gu, 2024). Recently, the *delta rule* has emerged as a popular design choice for linear attention and SSMs, enabling better experessivity and highly parallelizable implementations (Yang et al., 2024c). These architectural advances are complementary to our framework. While they focus on efficiently *processing* long contexts, TTT provides a mechanism for online *adaptation* to the information within that context. Our In-Place TTT can be also naturally integrated with these efficient backbones, as they also have MLP blocks. And we leave these as the future work.

**Memory Design and Augmentation.**    A related domain of research involves augmenting neural architectures with explicit memory modules to enhance their reasoning and contextual understanding capabilities. These approaches can be broadly distinguished by their function: some are designed to store persistent, task-agnostic knowledge in an external memory bank, while others focus on capturing transient, data-dependent information from the immediate context (Khandelwal et al., 2020; Guu et al., 2020; Lewis et al., 2020; Wang et al., 2024; Yu et al., 2025). The latter, contextual memories, have been implemented using various mechanisms, including recurrent state transitions, attention-based context aggregation in Transformers (Dai et al., 2019), and rapid, gradient-based updates to fast weights (Yang et al., 2024c). Test-Time Training (TTT) represents a powerful instance of this latter category, which conceptually extends the notion of a hidden state found in Recurrent Neural Networks (RNNs). Rather than compressing contextual history into a fixed-size activation vector, TTT designates a subset of the model's own parameters—the *fast weights*—to function as a high-capacity, dynamic memory (Ba et al., 2016; Schlag et al., 2021). These weights are updated on the fly at inference time, allowing the model to continuously internalize evolving contextual information and thereby function as an expressive, online evolving state.

## B    PROOF OF THEOREM 1

For completeness, we first restate the theorem with the precise bounds derived from the assumptions.

**Theorem 1** (Logit-wise Effect of LM-Aligned Target v.s. Reconstruction Target (Restated)). *Under the specified setting and assumptions, for a learning rate $\lambda_{lr} > 0$, the expected change in logits $\Delta \ell_n$*

*after one update step using the **NTP-aligned target** satisfies:*

$$\textbf{(Correct logit increases)} \quad \mathbb{E}\left[\Delta\boldsymbol{\ell}_n[v^*]\right] \geq \lambda_{lr} \cdot c_{norm}^2 \cdot c_{align}, \tag{4}$$

$$\textbf{(Other logits almost unchanged)} \quad |\mathbb{E}\left[\Delta\boldsymbol{\ell}_n[w]\right]| \leq \lambda_{lr} \cdot \epsilon \cdot c_{align}, \quad \forall w \neq v^*. \tag{5}$$

*In contrast, for the **reconstruction target**, the expected change in logits is negligible for the correct token:*

$$|\mathbb{E}\left[\Delta\boldsymbol{\ell}_n[v^*]\right]| \leq \lambda_{lr} \cdot \epsilon \cdot c_{align}. \tag{6}$$

*Proof.* We begin from the setup defined in Section 3.3. The change to the fast weights $\mathbf{W}_{\text{down}}$ from all prior context tokens is given by $\Delta\mathbf{W}_{\text{down}} = \lambda_{\text{lr}} \sum_{t\in\text{prior}} \mathbf{v}_t\mathbf{z}_t^\top$, where we use $\lambda_{\text{lr}}$ to denote the learning rate $\eta$ for consistency with the theorem statement. The resulting change in the logit for an arbitrary token $w$ at the query position $n$ is:

$$\Delta\boldsymbol{\ell}_n[w] = \mathbf{e}_w^\top(\Delta\mathbf{W}_{\text{down}})\mathbf{z}_n = \lambda_{\text{lr}} \sum_{t\in\text{prior}} \mathbf{e}_w^\top(\mathbf{v}_t\mathbf{z}_t^\top)\mathbf{z}_n. \tag{7}$$

Since $\mathbf{e}_w^\top\mathbf{v}_t$ and $\mathbf{z}_t^\top\mathbf{z}_n$ are scalars, we can rearrange the terms to get:

$$\Delta\boldsymbol{\ell}_n[w] = \lambda_{\text{lr}} \sum_{t\in\text{prior}} (\mathbf{e}_w^\top\mathbf{v}_t)(\mathbf{z}_t^\top\mathbf{z}_n). \tag{8}$$

To analyze the expected change, we take the expectation over the representations. Applying the linearity of expectation, we have:

$$\mathbb{E}[\Delta\boldsymbol{\ell}_n[w]] = \lambda_{\text{lr}} \sum_{t\in\text{prior}} \mathbb{E}[(\mathbf{e}_w^\top\mathbf{v}_t)(\mathbf{z}_t^\top\mathbf{z}_n)]. \tag{9}$$

Per our setup, the target vectors $\mathbf{v}_t$ (e.g., $\mathbf{e}_{x_t}$ or $\mathbf{e}_{x_{t+1}}$) are treated as determined. Thus, we can factor them out of the expectation:

$$\mathbb{E}[\Delta\boldsymbol{\ell}_n[w]] = \lambda_{\text{lr}} \sum_{t\in\text{prior}} \mathbf{e}_w^\top\mathbb{E}[\mathbf{v}_t\mathbf{z}_t^\top\mathbf{z}_n]. \tag{10}$$

Now, we invoke Assumption 2. It states that for the unique key position $t^*$, we have $\mathbb{E}[\mathbf{z}_{t^*}^\top\mathbf{z}_n] = c_{\text{align}}$, and for all other prior positions $t \neq t^*$, the updates provide no information gain, which implies $\mathbb{E}[\mathbf{v}_t\mathbf{z}_t^\top\mathbf{z}_n] = \mathbf{0}$. This simplifies the summation to a single term corresponding to the key-value pair $(k^*, v^*)$ at position $t^*$:

$$\mathbb{E}\left[\Delta\boldsymbol{\ell}_n[w]\right] = \lambda_{\text{lr}} \cdot \mathbb{E}\left[(\mathbf{e}_w^\top\mathbf{v}_{t^*}) \cdot (\mathbf{z}_{t^*}^\top\mathbf{z}_n)\right]. \tag{11}$$

We now analyze this simplified expression for the two target choices.

**Case 1: NTP-Aligned Target ($\mathbf{v}_{t^*} = \mathbf{e}_{x_{t^*+1}}$)**

First, we consider the logit of the correct token, $w = v^*$. Substituting the target and the token into Equation (11) yields:

$$\mathbb{E}\left[\Delta\boldsymbol{\ell}_n[v^*]\right] = \lambda_{\text{lr}} \cdot \mathbb{E}\left[(\mathbf{e}_{v^*}^\top\mathbf{e}_{v^*}) \cdot (\mathbf{z}_{t^*}^\top\mathbf{z}_n)\right]. \tag{12}$$

By **Assumption 1**, token embeddings have a non-trivial magnitude, $\|\mathbf{e}_{v^*}\|^2 \geq c_{\text{norm}}^2$. This gives us the lower bound in Equation (4):

$$\mathbb{E}\left[\Delta\boldsymbol{\ell}_n[v^*]\right] \geq \lambda_{\text{lr}} \cdot c_{\text{align}} \cdot c_{\text{norm}}^2. \tag{13}$$

Next, for any incorrect token $w \neq v^*$, the expected change is $\mathbb{E}\left[\Delta\boldsymbol{\ell}_n[w]\right] = \mathbb{E}\left[(\mathbf{e}_w^\top\mathbf{e}_{v^*}) \cdot (\mathbf{z}_{t^*}^\top\mathbf{z}_n)\right]$. Taking the absolute value and applying Assumption 1, which states that distinct embeddings are nearly orthogonal ($|\mathbf{e}_w^\top\mathbf{e}_{v^*}| \leq \epsilon$), we obtain the bound in Equation (5):

$$|\mathbb{E}\left[\Delta\boldsymbol{\ell}_n[w]\right]| \leq \lambda_{\text{lr}} \cdot c_{\text{align}} \cdot \epsilon. \tag{14}$$

**Case 2: Reconstruction Target ($\mathbf{v}_{t^*} = \mathbf{e}_{x_{t^*}}$)**

Here, we analyze the effect on the correct logit $w = v^*$. The expected change is:

$$\mathbb{E}\left[\Delta \boldsymbol{\ell}_n[v^*]\right] = \mathbb{E}\left[(\mathbf{e}_{k^*}^\top \mathbf{e}_{v^*}) \cdot (\mathbf{z}_{t^*}^\top \mathbf{z}_n)\right]. \tag{15}$$

In an induction task, the key $k^*$ is distinct from the value $v^*$. We again invoke Assumption 1 for these distinct tokens. Taking the absolute value gives the bound in Equation (6):

$$|\mathbb{E}\left[\Delta \boldsymbol{\ell}_n[v^*]\right]| \leq \lambda_{\mathrm{lr}} \cdot c_{\mathrm{align}} \cdot \epsilon. \tag{16}$$

This confirms that the reconstruction target has a negligible expected effect on the logit of the correct answer $v^*$.

The results from these two cases establish the claims in Theorem 1, providing a clear theoretical basis for the superiority of the NTP-aligned objective in the context of in-context learning. This completes the proof. □

## C  CONTEXT PARALLEL ALGORITHM FOR IN-PLACE TTT

For more clarity, we list the pseudocode of the context parallel implementation of our In-Place TTT here in Algorithm 1.

---

**Algorithm 1 In-Place TTT with Context Parallelism (Single Layer)**

---

**Require:** Pre-trained weights $\theta$ (incl. $W_{\mathrm{up}}, W_{\mathrm{gate}}, W_{\mathrm{down}}^{(0)}$); Conv1D kernel $K$; projection $W_{\mathrm{target}}$; learning rate $\eta$.

1: **Input:** Sequence chunks $\{X^{(i)}\}_{i=1}^T$.   ▷ Sequence partitioned for Context Parallelism (CP).
2: **for all** $i \in \{1, \ldots, T\}$ **in parallel do**       ▷ Step 1: Compute update deltas.
3:   $H_i \leftarrow \mathrm{AttentionBlock}(X^{(i)}; \theta)$     ▷ Standard attention, no changes required.
4:   $U_i, G_i \leftarrow H_i W_{\mathrm{up}}^\top, H_i W_{\mathrm{gate}}^\top$
5:   $Z_i \leftarrow \phi(G_i) \odot U_i$
6:   $V_i \leftarrow \mathrm{Conv1D}_K(X_0^{(i)}) W_{\mathrm{target}}$   ▷ Compute NTP-aligned target with causal padding.
7:   $\Delta W_i \leftarrow V_i^\top Z_i$        ▷ Compute gradient for the fast weight update.
8: **end for**
9: $\{S_i\}_{i=1}^T \leftarrow \mathrm{CUMSUM}(\{\Delta W_i\}_{i=1}^T)$     ▷ Step 2: Aggregate deltas associatively.
10: **for all** $i \in \{1, \ldots, T\}$ **in parallel do**   ▷ Step 3: Apply updates and compute outputs.
11:   $W_{\mathrm{down}}^{(i-1)} \leftarrow W_{\mathrm{down}}^{(0)} + \eta S_i$  ▷ Effective weight for chunk $i$ uses updates from chunks $< i$.
12:   $O_i \leftarrow Z_i (W_{\mathrm{down}}^{(i-1)})^\top$
13: **end for**
14: **At document boundaries:** Reset fast weights to $W_{\mathrm{down}}^{(0)}$.

---

## D  EXPERIMENT DETAILS

This appendix provides all details of the experimental settings, datasets, model configurations, and training hyperparameters used for the results presented in Section 4. The following subsections detail the setups for our three primary sets of experiments: the **continual pre-training** of Qwen3-4B-Base, the **from-scratch pre-training** of models at multiple scales (500M, 1.5B, and 4B), and the **targeted ablation studies**. Our goal is to provide sufficient detail to ensure the reproducibility of our findings.

### D.1  DETAILS OF DATASETS

For the large scale pretraining, continual pretraining, and ablation study, we use the dataset collected by ourselves, we give the details of these datasets as follows.

**From Scratch Pretraining Dataset.** The pretraining dataset mainly includes general English and Chinese text, along with high knowledge- or reasoning-density data, code, mathematics data, and

multilingual text, forming a balanced mixture of linguistic diversity, knowledge and reasoning-rich content, programming material, and mathematical reasoning.

**Continual Pretraining Dataset.** The continual pretraining dataset is designed to enhance long-context modeling: its short-document portion follows a distribution similar to Pretrain Data, while the long-document portion combines natural data such as books and repository-level code with synthetic data including retrieval-augmented and long-context-QA style constructions, ensuring both consistency with pretraining and coverage of challenging long-context scenarios. The data is organized into subsets with maximum sequence lengths of 32k and 128k for our two-stage training curriculum. During the continual pretraining experiments, we set the target value origin from the hidden states of the current layer, instead of the input embeddings. The convolution and projection are maintained to process the hidden states to get the target value.

## D.2 DETAILS OF TRAINING AND EVALUATION

**Training Details.** All models are trained on Nvidia H800 GPUs, with the detailed training hyperparameters listed in Tables 4 through 6.

Table 4: Training hyperparameters for 500M and 1.5B models.

| Hyperparameter | 500M Model | 1.5B Model |
|---|---|---|
| Optimizer | AdamW | AdamW |
| Learning Rate | 5e-4 | 3e-4 |
| Batch Size | 2M tokens | 4M tokens |
| Weight Decay | 0.1 | 0.1 |
| Gradient Clipping | 1.0 | 1.0 |
| Warmup Steps | 1024 | 1024 |
| Sequence Length | 32,768 | 32,768 |
| Tokens Trained | 20B | 60B |
| Sliding Window Size | 2,048 | 4,096 |

Table 5: Training hyperparameters for 1.7B models and 4B models pretraining

| Hyperparameter | value |
|---|---|
| Optimizer | AdamW |
| Learning Rate | 3e-4 |
| Batch Size | 8M tokens |
| Weight Decay | 0.1 |
| Gradient Clipping | 1.0 |
| Warm-up Tokens[1] | 1.6B |
| Sequence Length | 8,192 |
| Tokens Trained | 120B |

Table 6: Hyperparameters for two-stage continual pre-training.

| Hyperparameter | Stage 1 (32k Context) | Stage 2 (128k Context) |
|---|---|---|
| Base Model | Qwen3-4B-Base | Qwen3-4B-Base |
| Optimizer | AdamW | AdamW |
| Learning Rate | 5e-6 | 5e-6 |
| Weight Decay | 0.1 | 0.1 |
| Sequence Length | 32,768 | 131,072 |
| Tokens Trained | ~20B | ~15B |
| RoPE Extension | None | YaRN |
| Conv Size | 5 | 5 |

Table 7: Hyperparameters for continual pre-training of LLaMA-3.1-8B and Qwen3-14B-Base.

| Hyperparameter | LLaMA-3.1-8B | Qwen3-14B-Base |
|---|---|---|
| Optimizer | AdamW | AdamW |
| Learning Rate | 5e-6 | 5e-6 |
| Weight Decay | 0.1 | 0.1 |
| Sequence Length | 32,768 | 32,768 |
| Tokens Trained | $\sim$20B | $\sim$20B |
| RoPE Extension | None | None |
| Conv Size | 5 | 5 |

**Evaluation Details.** We employ the evaluation framework lm-evaluation-harness (Gao et al., 2024) to evaluate the models on the common sense reasoning benchmarks and employ the evaluation framework opencompass (Contributors, 2023) to evaluate the models on the long context benchmarks. All evaluation are conducted on Nvidia H800 GPUs.

In the evaluation of our continual pretrained Qwen3-4B model, we apply a clipping mechanism at inference time to ensure stable fast-weight updates. Specifically, if the Frobenius norm of an update delta $\|\Delta\mathbf{W}_{\text{down}}^{(i)}\|_F$ exceeds a predefined threshold $\tau$, the delta matrix is rescaled to have norm $\tau$ before being applied, i.e., $\Delta\mathbf{W}_{\text{down}}^{(i)} \leftarrow \tau \cdot \Delta\mathbf{W}_{\text{down}}^{(i)} / \|\Delta\mathbf{W}_{\text{down}}^{(i)}\|_F$. This prevents the accumulated updates from growing unboundedly as the sequence length increases, thereby maintaining numerical stability for long-context inference. For all reported evaluations of the Qwen3-4B model, this threshold was set to $\tau = 1e\text{-}5$.

To evaluate the efficiency of our In-Place TTT, we evaluate the prefill throughput and peak memory for sequence length ranging from 8k to 128k. We run the inference for our continual pretrained checkpoints based on Qwen3-4B-Base model. For the setting of sliding window, we set change the attention mechanism of these pretrained checkpoints to sliding window of 1024 tokens manually. We run the inference on Nvidia H800 GPUs with batch size of 1.

### D.3 DETAILS OF MODEL CONFIGURATION

This section details the architectural configurations of the models used in our experiments. All models are decoder-only Transformer architectures featuring standard components, including SwiGLU activations and Rotary Position Embeddings (RoPE) (Su et al., 2023). The key architectural parameters for all models trained from scratch are summarized in Table 8.

Table 8: Model architectural configurations for 500M and 1.5B Model.

| Parameter | 500M | 1.5B |
|---|---|---|
| Parameters (Approx.) | 500M | 1.5B |
| Hidden Size ($d_{\text{model}}$) | 1024 | 2048 |
| Num Layers | 24 | 24 |
| Num Attention Heads | 8 | 16 |
| FFN Hidden Size ($d_{\text{ff}}$) | 3072 | 6144 |
| Window Size | 2048 | 4096 |
| Vocabulary Size | 32,000 | 32,000 |
| Rope Base | 1e6 | 1e6 |

The models trained from scratch employ different attention mechanisms based on their experimental purpose. The 500M, 1.5B utilize sliding-window attention and we list the model configuration in Table 8. The 4B-scale experiments and ablation study evaluate two variants: one with full attention and another with sliding-window attention. The backbone architectures for the 4B models and 1.7B models are identical to the Qwen3-4B-Base model and the Qwen3-1.7B-Base model.

For the continual pre-training experiments described in Section 4.1, we start directly from publicly available pre-trained models—Qwen3-4B-Base (Yang et al., 2025), LLaMA-3.1-8B (Grattafiori et al., 2024), and Qwen3-14B-Base (Yang et al., 2025)—inheriting their architectures without modification.

In experiments featuring our method, the In-Place TTT module is integrated into the MLP blocks and applied to every sixth layer. For the ablation studies, this frequency is varied as described in the main paper. The training hyperparameters for Qwen3-4B-Base are listed in Table 6, and those for LLaMA-3.1-8B and Qwen3-14B-Base are listed in Table 7.

**Initialization of In-Place TTT Modules.** When integrating In-Place TTT into a pre-trained model for continual training, careful initialization is essential to preserve the model's pre-trained capabilities at the start of training. Specifically, we initialize the newly introduced TTT components—the Conv1D operator and the projection matrix $\mathbf{W}_{\text{target}}$—such that the TTT update $\Delta \mathbf{W}_{\text{down}}$ is negligible at initialization, ensuring the model begins from its original pre-trained behavior. Concretely, the depthwise Conv1D (kernel size 5, causal padding, no bias) is zero-initialized, so the target $\hat{\mathbf{V}}$ is zero at initialization. The projection matrix $\mathbf{W}_{\text{target}} \in \mathbb{R}^{d_{\text{model}} \times d_{\text{model}}}$ is initialized as a sparse diagonal matrix, where all off-diagonal entries are zero and the diagonal entries are drawn from $\mathcal{N}(0, \sigma^2)$ with $\sigma$ being the model's standard initializer range. This near-zero initialization of both components guarantees that the initial fast-weight update $\eta \hat{\mathbf{V}}_{[i]}^{\top} \mathbf{Z}_{[i]} \approx \mathbf{0}$, and consequently the effective $\mathbf{W}_{\text{down}}$ remains identical to its pre-trained value. As training progresses, the Conv1D and projection gradually learn to produce meaningful NTP-aligned targets, allowing the TTT mechanism to smoothly emerge without disrupting the pre-trained model.

# E  USAGE OF LLMS

During the preparation of this manuscript, LLMs was used to check grammar and improve readability. All authors have reviewed, edited, and take full responsibility for the paper's final version.

