# OpenReview forum: "In-Place Test-Time Training"
_ICLR.cc/2026/Conference — ICLR 2026 Oral_

### Official Review · Reviewer_cw7y · 2025-10-17

**Soundness:** 3
**Presentation:** 3
**Contribution:** 3
**Rating:** 6
**Confidence:** 3

**Summary:**

In this paper, the authors propose a new method that allows small LLMs to adapt and learn from new information while they are being used, rather than only during initial training. The approach, called In-Place Test-Time Training, updates specific internal components of the model efficiently without needing full retraining. Concretely, it uses the last layer of MLPs in transformer blocks. Further, it also introduces a new learning goal aligned with the model’s main prediction task and uses an optimized update process for speed and scalability. The authors show some experiments in small to moderate size LLMs showing that this method improves performance on long and complex tasks while remaining computationally efficient. Finally, they present a set of nice ablation studies.

**Strengths:**

This is a very good paper. In particular, I liked the following aspects of the paper:

1) The idea of the paper makes perfect sense, and the problem it tackles is important. Having LLMs that can be adapted at test time, improving their performance, is definitely a step in the right direction towards building strong models.

2) The authors solve the method using two simple key ideas: (1) using only the last linear layers in MLPs of Transformer blocks, (2) using a novel objective that somehow aligns with the task of next-token predictions. In both cases, these idea make complete sense, and are shown to be effective, despite being quite simple.

3) Because, the authors are using only a small subset of parameters, I assume that the method is quite computationally efficient.

4) As shown in several experiments, the method seems to improve over the baseline.

5) The paper is excellently written. I had a good time reading and understanding the paper.

**Weaknesses:**

1) The main weakness of the paper is that it has been evaluated in a relatively limited setting. Qwen3-4B, while a strong baseline, is still a very small LLM for modern standards, thus it is unclear if the method shows the same effectiveness when used with larger, or even significantly larger LLMs, which tend to be state-of-the-art.

2) Furthermore, it would have been cool if the authors would have checked the effectiveness of their method in other families of LLMs (even if having the same size), for example LLama, Mistral, Intern etc.

3) Finally, would the results hold if going in the domain of VLMs. Such results be it on Qwen3-VL or some version of InternVL would make the contribution of the paper even more solid.

**Questions:**

It would be nice if the authors are able to show those experiments. I would increase my score in such a case.

---

> ### Author Response · Authors · 2025-11-22
>
> We sincerely thank the reviewer for this valuable feedback. We agree that demonstrating the effectiveness of our method across larger scales and diverse model families can further strengthen our work.
>
> ## Regarding larger model size and other model families
>
> Thanks for your questions. In fact, we have also been investigating this direction after submitting our paper by conducting new experiments on the Qwen3-14B Base model. Besides, we further verify our approach on the Llama-3.1 Base model family during the author-reviewer discussion stage. The experimental settings are as follows:
>
> 1. Scalability on larger models (addressing weakness 1): adhering to the protocol in our submission, we conducted continual training on the Qwen3-14B Base model for about 15B tokens with a 32k context length and evaluated the model on the RULER benchmark to assess both long-context performance and extrapolation capabilities.
>
> 2. Versatility to other architectures (addressing weakness 2): to verify that our approach is architecture-agnostic, we extended our evaluation to the Llama-3.1-8B Base Model. Using the same training budget and context window, we also evaluated the model on the RULER benchmark.
>
> Experimental results are summarized in the table below, which indeed shows that our In-Place TTT approach consistently boosts the baseline's performance across different model architectures and scales.
>
> **LLAMA3.1-8B**
>
> |          | RULER-4K | RULER-8K | RULER-16K | RULER-32K | RULER-64K |
> | :------- | :------: | :------: | :-------: | :-------: | :-------: |
> | Baseline |   93.9   |   92.1   |   92.5    |   91.1    |   81.6    |
> | Baseline + IP-TTT   | **94.4** | **93.0** | **93.3** | **91.7** | **83.7** |
>
> **Qwen3-14B**
>
> |          | RULER-4K | RULER-8K | RULER-16K | RULER-32K | RULER-64K | RULER-64K+YarN |
> | :------- | :------: | :------: | :-------: | :-------: | :-------: | :------------: |
> | Baseline |   96.8   |   95.0   |   94.6    |   90.7    |   67.9    |      81.3      |
> | Baseline + IP-TTT   | **97.2** | **95.7** | **95.2** | **91.2** | **70.6** |    **82.5** |
>
> Note that because Llama-3.1 models have employed its customized RoPE frequency scaling strategy for context extension, the YaRN method is not applicable. These results strongly suggest that our In-Place TTT is a robust and scalable enhancement, delivering consistent gains regardless of model size or underlying architecture. We will update our manuscript to include these broader evaluations.
>
>
> ## Regarding VLM
>
> We appreciate the reviewer’s insightful suggestion regarding the applicability of our method to VLMs (e.g., Qwen-VL, InternVL).
>
> Our current submission focuses on establishing the theoretical and empirical foundations of In-Place TTT within the language modeling domain. LLMs evaluated in our paper indeed serve as necessary components for most state-of-the-art VLMs. Given that we have demonstrated robust improvements on both Qwen-3 and Llama-3.1 models (as detailed in above updated results), and considering that VLMs inherit the context-processing characteristics of their LLM backbones, we believe that our findings will generalize effectively to the multi-modal setting. We consider the adaptation of In-Place TTT to VLMs as a promising and critical avenue for future research to further broaden the scope of this technique.
>
> ---
> Thank you once again for your time and effort. We believe these experimental results can greatly strengthen our paper, and we would be grateful for your reconsideration of the overall evaluation.

---

### Official Review · Reviewer_LqGp · 2025-10-22

**Soundness:** 3
**Presentation:** 3
**Contribution:** 3
**Rating:** 8
**Confidence:** 4

**Summary:**

This paper introduces a Test-Time Training framework for LLMs which does not require any additional layers / parameters and can be introduced in existing pre-trained LLMS with some training as well as when a model is being trained from scratch. The main idea is to repurpose the down projection component of a GLU styled MLP into fast weights. Compared to previous fast-weight and TTT implementations, they also introduce a separate objective for the fast-weights which is more in tune with the overall LLM task of next-token prediction and is also theoretically motivated. Based on these two core ideas, the authors perform extensive evaluation of their technique, called In-Place TTT, on pretrained LLMs using continual learning as well as training LLMs of different sizes from scratch. They compare IPTTT to full attention, sliding window based attention methods and demonstrate the effectiveness of their scheme.

**Strengths:**

In my opinion, the core strength of this paper lies in the idea of repurposing the existing weight matrices in a neural network and using them as fast weights. This means their technique can be used to improve context management in existing LLMs as well as train large scale models using TTT. Moreover, the technique uses TTT not at the attention level but at the MLP block level which allows for efficient parallelization of the IPTTT framework using prefix sums and chunking sequences into token blocks. The theoretical analysis seems grounded and correct although I did not verify it fully. The experimental section is also done well and I don’t have any major concerns. The performance difference between their technique and the baseline, especially on long context tasks shows the promise of this technique as well. I believe this work will be useful to the community to build upon.

**Weaknesses:**

The main issue I have with the paper is certain sections confused me a little bit. There were some language issues, typos, and a few instances of confusion caused by notation. Here are some typos / language improvements I would suggest.

Line 161 -  also viewed  → also be viewed

Line 192 -  onece → once

Line 175 - we adapt only the MLP blocks → we only adapt the MLP blocks

Also, in the paragraphs on Efficient Adaptation with Chunk-Wise Updates, the target is referred to as V but then later it becomes V^{hat}. Is there a reason for that?

Also, I believe the authors have made bit of an overstatement in the explanation of  barrier (ii) "the canonical per-token update mechanism of TTT is inherently sequential, severely bottlenecking the parallel processing capabilities of modern accelerators.
And also here : "This chunk-wise update strategy is designed for modern hardware. By processing large blocks of
tokens at onece, it highly leverages parallelism and utilizes the computational power of GPUs or
TPUs, thus resolving the efficiency bottleneck that hinders prior research."

While it is true that Fast-weight like implementations are inherently sequential, chunk-wise training is not a new approach and was already explored / discussed in several fast-weight like papers e.g [1], [2]. I suggest the authors adjust for this.


[1] Titans: Learning to Memorize at Test Time

[2] : Fast weight programming and linear transformers: from machine learning to neurobiology

**Questions:**

The major confusion I have about the methodology is understanding how it works during inference. In regular fast weights, the goal is to make sure the fast weights compress the context (the reconstruction objective). But here, it is to have them predict info about the future tokens using V. However, V for future tokens is not available during inference, so how do the fast weights even updated?

---

> ### Author Response · Authors · 2025-11-22
> **Official Comment by Authors (Part 1)**
>
> We sincerely thank the reviewer for the thorough evaluation and constructive feedback. We appreciate that you found our "In-Place" framework promising. Your comments regarding clarity, notation, and the inference mechanism help us improve the manuscript. We have addressed all the raised points in our revised submission, as detailed below:
>
> ## Regarding the clarifications on language, typos, and notations
>
> Thanks for your careful reading. We apologize for the oversight regarding the language errors and typos (e.g., Line 161, 175, 192). We have corrected all specific instances you identified and have conducted a comprehensive proofread of the entire manuscript again to ensure high standards of readability.
>
> Regarding the notation confusion between $V$ and $\hat{V}$ in Section 3.1, we appreciate you pointing this out. In our original draft, we used $V$ to denote a generic value target and $\hat{V}$ (in Equation 1) to denote the specific, constructed target derived from our LM-aligned objective, which helps readers to distinguish the different meanings. In the revised version of our paper, we have unified the notation and added an explicit definition at the first occurrence of the target variable to ensure mathematical consistency throughout the Methodology section.
>
> ## Regarding Chunk-Wise Updates and Related Work
>
> Thank you for this point and providing these highly relevant references ([1, 2]). We would like to clarify that it is not our intention to mis-credit the invention of chunk-wise update mechanism. In our submission, we would like to contrast our approach specifically with the canonical per-token update prevalent in recent TTT-for-LLM works, which creates efficiency bottlenecks on GPUs. We also acknowledged that the concept of chunk-wise/block-wise processing exists in the broader fast-weight and TTT literature.
>
> Firstly, we carefully include prior works that explore the usage of chunk-wise updating approaches in sequence modeling [3-5], which extend your mentioned two works and would be a good supplementary for us to discuss and credit. Furthermore, we carefully follow your suggestions to revise our submission to ensure correct, objective, and clarified statements:
>
> - Credit to prior works that explore chunk-wise updating approaches: we explicitly cite and carefully credit prior works in all related paragraphs involving our chunk-wise mechanism, including your mentioned ones, to avoid misunderstanding of the literature.
>
> - Clarification of contributions: we have also refined our claims to clarify that our contribution is not the invention of chunking itself, but rather the efficient application of large-chunk-wise updates within an in-place, "drop-in" MLP adaptation framework. Please refer to lines 058-062, 073-077, 185-194, and 208-211, we explicitly emphasize that if a standalone TTT layer is used to replace the attention mechanism, maintaining high-quality local information for every token often necessitates fine-grained, per-token updates, making large-chunk-wise updates for acceleration not viable due to the performance constraints. However, in our In-Place framework, the retention of the attention mechanism liberates us from this constraint. This allows us to update the MLP using large chunks—fully leveraging the parallelization benefits of chunk-wise processing found in recent works—without compromising local context modeling. Moreover, Our experiments also demonstrate that our framework is **naturally well-suited for chunk-wise—and specifically large chunk-wise—updates**. In our ablation study (Section 4.3), for a total sequence length of 8k, larger chunk sizes of 512 and 1024 achieve optimal performance. Thus, our adoption of chunk-wise updates serves not only as an efficiency optimization but also as a key driver for superior model performance.
>
> - Expanded related works: We have further updated the Related Work Section (Section A) to explicitly discuss these works [1-5]. We credit these works for exploring chunk-wise fast weight updates while highlighting how In-Place TTT differentiates itself by repurposing existing Transformer weights and optimizing for the auto-regressive Next-Token Prediction (NTP) objective without retraining from scratch.
>
> The draft of our submission has also been updated along these responses. Combining all these modifications, we hope the readers can correctly understand the literature and the contributions in our work.
>
> [1] Titans: Learning to Memorize at Test Time.
> [2] Fast weight programming and linear transformers: from machine learning to neurobiology.
> [3] Sequential-Parallel Duality in Prefix-Scannable Models.
> [4] Retentive Network: A Successor to Transformer for Large Language Models.
> [5] Test-time training done right.

---

> ### Author Response · Authors · 2025-11-22
> **Official Comment by Authors (Part 2)**
>
> ## Regarding the Inference Process
>
> In fact, our method maintains strict causality during inference. The fast weights are never updated using "future" tokens that have not yet been generated.
>
> The inference process operates as follows:
>
> 1. Generation (Chunk $k$): The model generates a chunk of tokens (Chunk $k$) using the current fast weights $W^{(k)}_{down}$.
> 2. Target Construction: Once Chunk $k$ is fully generated (or fully received, in the case of a prompt), we compute the keys $Z_{[k]}$ and the targets $\hat{V}_{[k]}$.
>     * Crucially: Although $\hat{V}$ contains "future" information relative to a specific token $t$ (e.g., $x_{t+1}$), this information is derived entirely from within the already generated Chunk $k$.
>     * Boundary Handling: For tokens at the very end of Chunk $k$, where a convolution might require data from $k+1$, we apply padding strategy (as mentioned in Section 3.4). This forces the target to rely only on available history.
> 3. Update: We perform the gradient update to obtain $W^{(k+1)}_{down}$.
> 4. Next Step: The updated weights are used to generate Chunk $k+1$.
>
> This "Generate $\rightarrow$ Update $\rightarrow$ Generate Next" loop ensures that we only ever train on data that the model has already seen, making the inference process identical to training and strictly causal. We hope these clarifications and revisions satisfactorily address your confusing.
>
> ---
>
> Thank you once again for your time and effort. We believe these revisions can further strengthen our submission. We are willing to respond to any more questions you may have.

---

### Official Review · Reviewer_TXQK · 2025-11-01

**Soundness:** 3
**Presentation:** 4
**Contribution:** 3
**Rating:** 8
**Confidence:** 3

**Summary:**

The paper introduces a novel in place test time training framework that enables LLMs to dynamically adapt a set of ‘fast’ weights during inference time with new information. The authors provide a way to repurpose the existing final projection matrix of the MLP blocks of the decoder transformers and avoid costly retraining. They also provide an efficient chunk wise update rule and an objective that aligns with the next token prediction task. Lastly they also provide a context parallel algorithm. Experiments on a 4B qwen show that the approach achieves improvements over baseline. Good practical paper, just a few questions below.

**Strengths:**

1. The method is practical and drop in design that is novel and straightforward to integrate in practical deployments
2. The authors have made the learning objective aligned with next token prediction from a generic reconstruction objective and show ablations to prove their point
3. The authors also additionally provide context parallel algorithms for maximal computation efficiency

**Weaknesses:**

1. The experimental section is based on a single family of model qwen. What if selecting that specific layer is suboptimal in some other family of models like llama. What will the authors do with a mixture of experts paradigm?

2. The authors provide limited justification for selecting the MLP block’s final projection matrix. There are other MLP blocks with large matrices inside the decoder transformer as well, perhaps the attention layers output projection matrix etc. This seems like a heuristic and an ablation could make it clear which layer is the optimal layer to choose for this purpose. What if a combination of layers provides the best performance.

3. On that note isn't it possible to do a modified Lora TTT[1]? For example, instead of changing the weights of the model itself directly as fast weights, we keep a set of lora weights that we keep updating at inference time, and in lieu of its small flops footprint, instead of changing the full matrix of only the projection matrix, one can now change a few different matrices across the transformer block in a flop matched regime. Isn't that a reasonable baseline to consider?

References:
1. Kojima, Yuto, Jiarui Xu, Xueyan Zou, and Xiaolong Wang. "Lora-ttt: Low-rank test-time training for vision-language models." arXiv preprint arXiv:2502.02069 (2025).


3. Some figures like figure 3 are missing legends.

**Questions:**

see weaknesses

---

> ### Author Response · Authors · 2025-11-22
> **Official Comment by Authors (Part 1)**
>
> We sincerely thank the reviewer for the constructive feedback and recognition of our work. We appreciate the insightful questions, which have allowed us to strengthen the paper's empirical foundation and scope. We address each point below.
>
> ## Regarding models in other families
>
> Good point! Following your suggestion, we further verify our approach by conducting experiments on the **Llama-3.1-8B Base** model. Following the protocol detailed in our submission, we performed continual training for 15B tokens with a 32k context length. We then evaluated the trained model on the RULER benchmark to assess both long-context retention and extrapolation capabilities.
>
> As summarized in the table below, our In-Place TTT consistently boosts the baseline performance with various context lengths. Notably, we observe substantial gains on the RULER-64k extrapolation setting—doubling the training context—which confirms that our approach functions as a robust, architecture-agnostic enhancement.
>
> |          | RULER-4K | RULER-8K | RULER-16K | RULER-32K | RULER-64K |
> | :------- | :------: | :------: | :-------: | :-------: | :-------: |
> | Llama-3.1-8B Base |   93.9   |   92.1   |   92.5    |   91.1    |   81.6    |
> | Llama-3.1-8B Base + IP-TTT   | **94.4** | **93.0** | **93.3** | **91.7** | **83.7** |
>
>
> ## Regarding the MoE paradigm
>
> We thank the reviewer for highlighting the connection to the Mixture-of-Expert paradigm. In general, our In-Place TTT framework is inherently modular and can indeed be adapted to the MoE paradigm in two compelling ways:
>
> - Direct Application (Shared Expert): Many open-source MoE architectures (e.g., DeepSeek-MoE, Qwen-MoE) utilize a "shared expert" that processes every token alongside routed experts. The most straightforward and efficient application of our method is to treat the final projection matrix of this shared expert as the fast weight. This implementation mirrors our current dense MLP setting, requires no new custom operators, and allows for seamless integration. To empirically validate this, we are currently pre-training a 7B MoE model (0.7B active parameters) from scratch using this shared-expert adaptation. We will update these results during the author-reviewer discussion stage as soon as possible.
>
> - All-Expert Adaptation: A more advanced adaptation involves applying our In-Place TTT to the projection matrices of all experts. This would dramatically increase the adaptive "state size" of the fast weights, potentially offering a much higher ceiling for models processing and reasoning over long-horizon, evolving tasks. We present our solutions for this setting here: formally, let $x_i$ be the $i$-th token input, $S_i$  be the set of top-k selected experts, and $p_{i,e}$ be the routing weight for expert $e \in S_i$. The MoE module calculates $W_{down}^{(e)}\left(\phi(W_{gate}^{(e)}x_i) \odot (W_{up}^{(e)}x_i)\right)$, where we define the intermediate activation for expert $e$ as $h_{i,e} = \phi(W_{gate}^{(e)}x_i) \odot (W_{up}^{(e)}x_i)$. To adapt the $W_{down}^{(e)}$ as fast weights, we maximize a dot-product objective $\mathcal{J}$ between the outputs of MoE layer and the TTT target $v_i$:
>
> $$J = \sum_{i=1}^{L} \sum_{e \in S_i} p_{i,e} \langle v_i, W_{down}^{(e)} h_{i,e} \rangle$$
>
> Consequently, the fast weight update $\Delta W_{down}^{(e)}$ for each expert $e$ is derived from the gradient of this objective, aggregating only the tokens routed to it:
>
> $$\Delta W_{down}^{(e)} = \eta \nabla_{W_{down}^{(e)}} \mathcal{J} = \eta \sum_{i: e \in S_i} p_{i,e} v_i h_{i,e}^\top$$
>
> Efficiently implementing this module thus requires specialized high-performance operators to handle the dynamic, unbalanced workloads across experts:
>
> * Sparse Gather & Scatter Operators: operators that efficiently "group" non-contiguous tokens routed to a specific expert into a contiguous block for processing, and subsequently "restore" the computed results back to their original in-sequence order.
> * Fused Parallel Kernels: operators that parallel apply the TTT updates for all experts simultaneously in a single GPU kernel launch. Without this, the system would suffer from the massive overhead of launching many tiny, sequential operations for different experts due to dynamic and unbalanced workloads across experts.
>
> All of these operators can be efficiently implemented using modern programming languages such as CUDA or Triton. This is indeed a promising direction, and we will investigate it in the future due to bandwidth limits. Thank you for this point and we will carefully include this discussion in our revision.

---

> ### Author Response · Authors · 2025-11-22
> **Official Comment by Authors (Part 2)**
>
> ## Regarding the choice of Fast Weights
>
> Thank you for this point. It is indeed true that our In-Place TTT approach can be also applied to other linear projection matrix in the Transformer architecture. In our submission, the selection of the MLP's final projection matrix ($W_{down}$) was primarily motivated by its role as one of the largest parameter matrices in a Transformer block. We hypothesized that fast weights with larger "state sizes" would provide greater adaptive capacity—a hypothesis supported by our "state size" ablation study (Figure 3a). We acknowledge the reviewer's point that other large matrices (e.g., attention output projection $W_O$, but smaller than $W_{down}$) or combinations of layers are plausible alternatives. However, a systematic, exhaustive comparison of all possible "in-place" candidates would be a significant study in its own right. Our primary goal in this work is to demonstrate the viability and effectiveness of the "in-place" paradigm itself, for which the $W_{down}$ matrix serves as a strong, well-motivated baseline. We agree that exploring optimal layer combinations is a valuable research question for future work.
>
>
> ## Regarding Low-Rank TTT
>
> We thank the reviewer for this insightful suggestion. We agree that a LoRA-based TTT formulation—updating low-rank adapters instead of full weights—is a highly practical direction that could significantly reduce computational costs and memory overhead during inference.
>
> Inspired by your comment, we conducted exploratory experiments during the discussion stage to assess the potential of this approach. Specifically, we analyzed the intrinsic properties of the weight update matrix $\Delta W$ generated by our method using the 4B parameter model trained from scratch in our submission. We evaluated the effective rank of the update matrix $\Delta W$ on the ArXiv and Github subsets of The Pile dataset by measuring the relative Frobenius norm error when approximating the original update with low-rank matrices $\Delta W^{\text{Rank } R}$ via Singular Value Decomposition:
>
> $$\text{Relative Error} = \frac{\|\Delta W^{\text{Rank } R} - \Delta W\|_F^2}{\|\Delta W\|_F^2}$$
>
> The approximation errors at different ranks are summarized below:
>
> | Layer | Dataset | Rank=32 | Rank=64 | Rank=128 | Rank=256 |
> | :---: | :--- | :---: | :---: | :---: | :---: |
> | **0** | **ArXiv** | 0.052497 | 0.025186 | 0.008768 | 0.001330 |
> |       | **Github** | 0.031240 | 0.011526 | 0.002872 | 0.000258 |
> | **18** | **ArXiv** | 0.097831 | 0.058860 | 0.028936 | 0.009387 |
> |       | **Github** | 0.058533 | 0.028588 | 0.011740 | 0.003222 |
> | **30** | **ArXiv** | 0.119595 | 0.071104 | 0.032369 | 0.009318 |
> |       | **Github** | 0.071691 | 0.037414 | 0.014811 | 0.003345 |
>
>
> Notably, the original $\Delta W$ has dimensions of $9728 \times 2560$. The above results demonstrate low approximation errors at ranks between 64 and 256, indicating that the rank can be effectively reduced by a factor of approximately 10 to 40 times. These results indicate that the update matrix $\Delta W$ produced by In-Place TTT is relatively sparse and possesses a low effective rank. This empirical evidence suggests that the fast weights can indeed be efficiently approximated or parameterized using low-rank structures without significant loss of information.
>
> While there are diverse ways to parameterize and train such low-rank updates, and determining the optimal configuration , these findings confirm that our framework can likely be extended to low-rank regimes for even greater efficiency. We view this as a compelling avenue for future work.
>
>
> ---
> Thank you again for your thoughtful feedback. We will incorporate these results and discussions into the final version of our paper, and we would be happy to answer any further questions.

---

> > ### Comment · Reviewer_TXQK · 2025-11-25
> >
> > I really appreciate the authors to have gone out of their way to add new experimental results even though it was not explicitly asked for. This additional evidence(s) regarding the moe and the low rank directions significantly strengthens the paper's overall findings and clarity would indeed make the paper stronger if added in the appendices at the very least, but again is not necessary as it stands currently. Your explanation of the MoE in the shared expert paradigm is what I expected. The non-shared expert regime is I agree, also a bit of a tough one since it may additionally have non trivial impact on the routing logic across different layers.
> >
> > My score remains at 8, as this reflects my final assessment of the work based on its core contribution.

---

> > > ### Author Response · Authors · 2025-11-25
> > >
> > > Thank you for your thoughtful feedback and for acknowledging our additional analyses and experimental results. If you have further questions or suggestions at any stage, we would be happy to address them. Thank you again for your time and support.

---

### Author Response · Authors · 2025-12-03
**Summary of the Rebuttal Phase**

Dear Area Chair,

We thank you and the reviewers for the time and effort dedicated to reviewing our work. We are pleased that the reviewers have acknowledged the novelty and effectiveness of In-Place TTT. In particular, the reviewers highlighted:

* **Significance & Impact:** The reviewers strongly validated the paper's direction, acknowledge that "the idea of the paper makes perfect sense, and the problem it tackles is important." They emphasized that our approach to test-time adaptation is "definitely a step in the right direction towards building strong models"

* **Practicality ("Drop-in" Usability):** The reviewers commended our In-Place TTT framework for its "simple yet effective" design. They emphasized that the ability to treat projection matrices as adaptable fast weights allows for a "seamless integration" into existing LLMs, enabling dynamic adaptability without complex architectural modifications.

* **Theoretical Motivation & Efficiency:** The reviewers acknowledged our principled design choices, specifically noting that the Next-Token-Prediction objective is theoretically motivated and the chunk-wise update mechanism renders the framework highly efficient and scalable.


During the rebuttal period, we actively engaged with the reviewers to address their remaining questions, which primarily focused on the need to demonstrate effectiveness across larger scales and diverse model families.

To address this concern, we conducted comprehensive additional experiments by applying In-Place TTT to state-of-the-art open models, specifically Llama-3.1-8B and Qwen3-14B. We evaluated these models on the rigorous RULER benchmark across varying context lengths. These experiments demonstrate that our method generalizes effectively beyond the initial models presented, consistently boosting performance across different architectures and parameter scales.


Furthermore, Reviewer cw7y has explicitly stated that "It would be nice if the authors are able to show those experiments. I would increase my score in such a case." in the review. Therefore, we believe that we have addressed these concerns and the reviewer would like to raise the score.

We believe we have successfully resolved the key questions raised during the review process. And we are confident that this work makes a significant and timely contribution to the ICLR community.

Sincerely,

The Authors

---

### Public Comment · ~Shiteng_Lu1 · 2026-03-18
**Follow-up on Code Availability**

Hi Authors,
Thanks again for your great work and for mentioning that the code will be released to support reproducibility and future research. I truly appreciate your commitment to open science!
I’ve looked but haven’t been able to find the repository yet—could you kindly share an estimated timeline for when the code might become publicly available? Even a rough indication (e.g., “within a month”) would be very helpful for those of us hoping to build on your work.
Looking forward to it, and thanks in advance!

---

> ### Public Comment · ~Guhao_Feng1 · 2026-04-07
>
> Hi, thanks for your interest in our work!
>
> The code is now publicly available at: https://github.com/ByteDance-Seed/In-Place-TTT
>
> Feel free to reach out if you run into any issues or have questions. Happy to help!

---

### Meta-Review · Area_Chair_jQmK · 2025-12-22

**Summary:**

1. Limited backbones (sizes and model families): The experimental section is based on a single family of model Qwen3-4B. (Reviewer TXQK and cw7y)
2. The selection of the final projection matrix seems heuristic and needs an ablation study. (Reviewer TXQK)
3. This paper has some writing issues or typos. (Reviewer LqGp)
4. This paper lacks discussions about Chunk-Wise Updates and related work. (Reviewer LqGp)

**Reviewer Concerns:**

The authors have almost solved all concerns mentioned by the reviewers during the rebuttal. It is necessary to include these discussions in the final version and further refine the writing and typos to improve the paper.

**Reviewer Scores:**

Reviewer TXQK: 8

Reviewer LqGp:  8

Reviewer cw7y:  6 --> 8

---

### Decision · Program_Chairs · 2026-01-26

Accept (Oral)